# Influence of Hydraulics on Electric Drive Operational Characteristics in Pump-Controlled Actuators

**Viacheslav Zakharov** 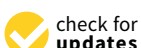 **and Tatiana Minav ***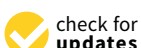

IHA—Innovative Hydraulics and Automation, ATME, Faculty of Engineering and Natural Sciences (ENS), Tampere University, Korkeakoulunkatu 6, 33720 Tampere, Finland; viacheslav.zakharov@tuni.fi
* Correspondence: tatiana.minav@tuni.fi

**Abstract:** Transitioning from a valve- to pump-controlled system has been observed in working hydraulics to reduce energy consumption and accelerate the response to decarbonization requirements in Non-road Mobile Machinery (NRMM). The utilization of an electric motor as a prime mover significantly enhances the chances of completing this task. While the concept of combining hydraulics with an electric motor is not new, the functionality of electro-hydraulics can be improved due to multidisciplinary domains, the impact of the electric drive and its control is usually underestimated. Thus, this study aims to evaluate the influence of hydraulics on electric drive operational characteristics in pump-controlled actuators. It utilizes an electro-hydraulic model and simulation study under various conditions, a stability analysis, and a Pulse Width Modulation (PWM) of the frequency converter (FC) evaluation on a selected variable speed pump-controlled actuator to demonstrate the influence of hydraulics on an electric drive. Experimental validation of the model demonstrated an acceptable accuracy of 5%. Moreover, the stability analysis indicated a rise time of about 0.051 s, an overshoot of 0.53%, a transient process time of 0.13 s, and a steady-state value difference of 0.16%; all of which guarantees stable operation of the electric drive with a hydraulic load. In addition, an optimal PWM of an FC frequency of 5 kHz was selected to guarantee accurate speed control with minimum overshoot.

**Keywords:** direct-driven hydraulics (DDH); pump-controlled actuators; field-oriented control (FOC); permanent magnet synchronous motor (PMSM); non-road mobile machinery (NRMM)

## 1. Introduction

As a continuation of the previous research in [1], this paper addresses the utilization of alternative rotational power sources in Non-road Mobile Machinery (NRMM), such as an Electric Motor (EM) for the powering of the working hydraulics instead of an Internal Combustion Engine (ICE). Electrification and hybridization of the working hydraulics is a means to support the decarbonization of NRMM. In terms of the latest trends, working hydraulics can be divided into two categories: valve- and pump-controlled systems [2]. Valve-controlled systems have a large variety of structures such as conventional load-sensing, digital hydraulics, independent metering, and common pressure rail. Load-sensing systems are characterized by quite hard falls, such as pump instability and pump-load instability; however, there are several solutions, such as electronic controllers, that can offer improvements [3] over the conventional low efficiency of 21% [4]. Digital hydraulic systems require a high number of valves, which allows a binary representation to adjust the flow according to the needs and to significantly reduce throttling losses [5]. A common pressure rail allows for an increase in efficiency in comparison with load-sensing systems; however, huge metering losses and an inability to recover energy are regarded as a disadvantage from a long-run perspective [6]. Independent metering also demonstrates less energy consumption than load sensing systems but requires constant system status monitoring [7]. As an alternative to valve-controlled systems, pump-controlled actuators can be

utilized to improve system efficiency. Pump-controlled systems can be categorized as variable speed, variable displacement, or a combination of both. A system that combines speed and displacement controls are highly challenging, due to the complex control structure of such a system with predictive observers [8]. Examples of single pump and double pump solutions for variable-displacement and speed pump-controlled systems can be found in [2,9], respectively. Both of these pump solutions benefit an improvement in efficiency compared to conventional valve-controlled solutions. Most pump-controlled systems require the flow compensation approaches to balance the flow in a single-rod cylinder, which is the most common actuator in NRMM due to limited space requirements. An overview of the flow compensation valve-based solutions is detailed in [10]. Alternative solutions for flow compensation can be a double pump [11], an additional charging pump [12], or the utilization of the hydraulic transformer [13,14]. Commonly, these pump-controlled systems utilize EM as a prime mover. While the concept of combining hydraulics with an electric motor is not new, the functioning of electro-hydraulics is often underestimated and can be improved through multidisciplinary domains, as the impact of the electric drive and its control is usually neglected. As an example of a pump-controlled actuator, the authors evaluated the behavior of an electro-hydraulic system only in hydraulic terms [15]. Utilization of the speed signal instead of a complete mathematical model of EM is common and presented in [16]. However, the influence of the electric drive on the hydraulic side of the system may be significant. Control of the electric drive (e.g., speed or torque) impacts the flow of the pump in speed-variable pump-controlled systems [17,18]. The main parts of the electric drive, which have an influence on operation characteristics, are the EM and Frequency Converter (FC). Consequently, the parameters of the electric drive, such as the quality of feeding voltages and currents, have a direct impact on the behavior of the shaft of the motor (e.g., torque ripples, vibrations, and speed of reaction) and the system as a result [19]. Moreover, even voltages and currents have a direct relation with FC factors, such as the type of control system, number of FC levels, strategy of FC control, and frequency of Pulse Width Modulation (PWM). Thus, this study aims to evaluate the influence of hydraulics on electric drive operational characteristics in pump-controlled actuators on an example of a speed-variable system. As a part of this study, a stability analysis is performed to investigate the behavior of the system and the factors that affect it. In addition, a step response analysis demonstrated the main system stability characteristics, such as rise time, overshoot, vibrancy, transient process time, and steady-state values. The set of PWM frequencies was compared and the behavior of the system under the different FC frequencies was demonstrated to evaluate the influence of hydraulics on an electric drive. The accuracy of the simulated model was proven with validation in a laboratory test rig. The analysis demonstrated that the system has a stable behavior, and all transient processes are within the required limitations. Furthermore, a PWM analysis under hydraulic load demonstrated that the dependency of the behavior of the system on the PWM frequency is different compared to other types of load. Increasing the PWM frequency improves the accuracy of characteristics and the stability of the system due to a decrease in the noise produced by the frequency converter. This study demonstrated that increasing the PWM frequency by some value decreases the stability. This new phenomenon needs to be observed and analyzed further. The remainder of the paper is divided into five sections. Section 2 presents the system description with the principles of operation of the selected variable-speed pump-controlled actuator. This section is divided into two subsections: the selected test rig description and modeling. Section 3 contains the validation of the model. Section 4 describes an analysis of the simulation results. Sections 5 and 6 discuss the implications of these results and outline areas for future research.

## 2. System Description

Direct-driven Hydraulics (DDH) is one example of pump-controlled systems utilizing EM as a prime mover. DDH can be described as a highly efficient system with fast response and dynamics [20,21]. The structure of the DDH system is outlined in this section. A pump-

controlled system schematic, electric drive schematic, component tables, and working principle are defined in Section 2.1. The hydraulic system model and electric drive model are described in Section 2.2 with mathematical equations and control laws.

### 2.1. Test Rig Description

The DDH system is an electro-hydraulic system that converts electric energy into rotational mechanical energy and then to hydraulic form via the pump/motor. In this research, note that pump/motor refers to units that operate as a pump during the lifting operation and as a motor during the lowering operation. The hydraulic part of the system consists of two hydraulic pumps/motors, a hydraulic cylinder, a hydraulic accumulator (responsible for the pressure compensation), and an oil tank. The schematic of the DDH system is presented in Figure 1 below.

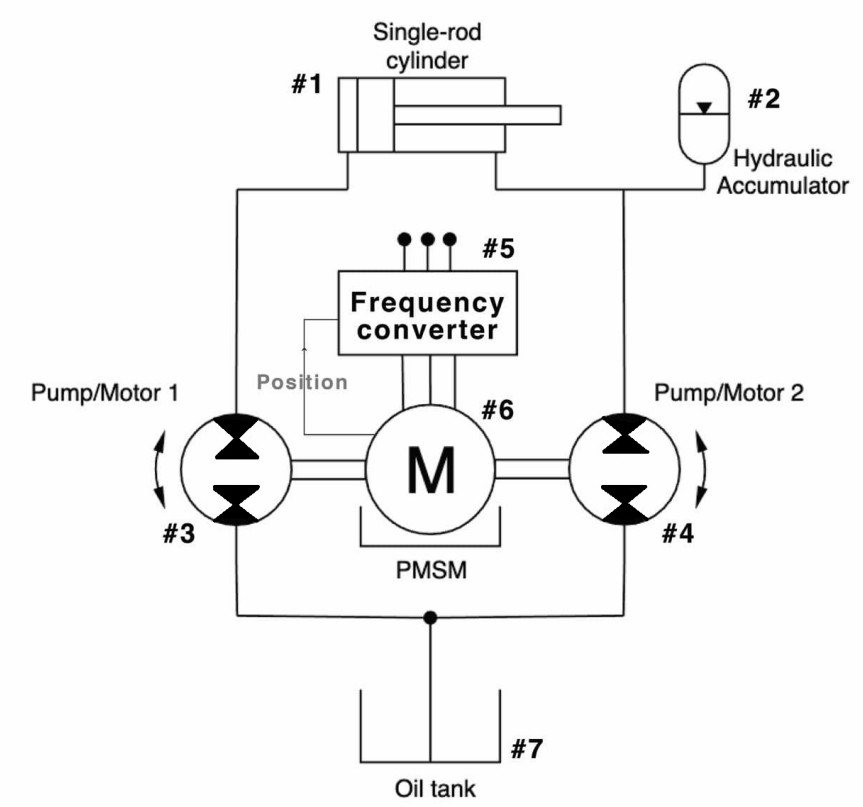

**Figure 1.** DDH drive. 1—Single-rod cylinder, 2—Hydraulic accumulator, 3—pump A, 4—pump B, 5—Frequency converter, 6—PMSM, 7—Oil tank.

A single-rod cylinder is used in this research. The utilization of a single-rod hydraulic cylinder is challenging because of the unbalanced input and output flows due to the different areas of the cylinder sides. As a result, it requires dimensioning of the pump/motor unit sizes and a control for the equable movement of the cylinder stroke [19]. The stroke position is employed as a feedback value for the whole system. The hydraulic accumulator (#2) allows for stabilizing pressure on the rod side of the system. The EM (#6) rotates the shaft of the pump/motor 1 (#3) and pump/motor 2 (#4). Both units are fixed-displacement hydraulic pumps/motors. The different displacements of pump/motor 1 (#3) and pump/motor 2 (#4) were selected to match the displacement of the cylinder (#1). The pump/motor unit creates a flow of hydraulic fluid (oil), which comes from the oil tank (#7) to the cylinder chambers. A list of the hydraulic components of the system is shown in Table 1 below.

**Table 1.** Parameters of components of DDH.

| #1 Hydr. Cylinder | #3 Hydr. Pump/Motor 1 | #4 Hydr. Pump/Motor 2 | #2 Hydr. Accumulator |
|---|---|---|---|
| MIRO C-10-60/30 × 400 | XV-2M/14 | XV-2M/22 | HAD0.7-250-1X/80G04A |
| Max pressure 190 bar | Max pressure 250 bar | Max pressure 200 bar | Max pressure 350 bar |
| Area A 0.0028 mm$^2$ | Disp. 14.4 cm$^3$/rev | Disp. 22.8 cm$^3$/rev | Precharge pressure 10 bar |
| Area B 0.0021 mm$^2$ | Vol. efficiency 97.22 % | Vol. efficiency 96.49 % | Volume 0.7 L |
| Stroke length 0.4 m | Max. rot. speed 3500 rpm | Max. rot. speed 3000 rpm | Type: Hydro-pneumatic |

The electric drive consists of a three-phase Permanent Magnet Synchronous Machine (PMSM) (#6), a rectifier, an inverter (both the rectifier and the inverter are known as frequency converters) (#5), and a control unit. A rectifier takes the three-phase AC and transforms it into DC. The rectified DC comes to the inverter, which converts it to three-phase AC with another amplitude and frequency. DC-to-AC converting is realized by alternately turning on and off transistors at each of the phases. The varying frequency and amplitude of the three-phase AC is the main control principle of adjustable electric drives. Emerson Control Techniques Unimotor FM115 U2C300 VACAA115190 is used as a prime mover. The main parameters of the EM are presented in Table 2 below.

**Table 2.** Parameters of PMSM.

| Rated/Peak Torque, [Nm] | Inertia, [kg cm$^2$] | Rated/Max Speed, [rpm] | Rated Current, [A] | Voltage, [VAC] |
|---|---|---|---|---|
| 9.4/37.6 | 9.0 | 3000/4800 | 5.9 | 380/480 |

The frequency converter (FC) is the main control part of the electric drive. The FC is modeled as a cellular structure of 12 transistors. Every transistor is controlled by separate Pulse Width Modulation (PWM) control signals. The Space Vector Modulation (SVM) was selected as the most utilized approach to influence the output currents and voltages. The principle of the SVM work is described in detail in [21]. The basis of the modulated signals is calculated by a control system.

Vector control systems allow for the regulation of the EM speed, even with a changing load on the shaft and at a wide range of speed [22]. The reference speed, known load torque level, and feedback from the EM (i.e., voltage, current, and shaft position) are used as system inputs. The feeding currents and voltages are the outputs of the system and are applied to the calculation of control signals of the FC transistors.

The operating principle of the entire electro-hydraulic system is based on the positional control (depicted in Figure 1). The electric drive uses the piston position and torque as input signals and feedback, and the rotational speed is utilized as input by the hydraulic part of the system. A closed-loop structure provides the possibility of stable regulation of the whole system.

*2.2. Modeling*

This subsection describes the mathematical equations of DDH and the structure of the control system. The variable rotational speed of the prime mover is used to change the flow produced by the pump/motor. In this research, it is a determinative variable of the flow equation. The pump/motor behavior is represented by Wilson's model.

$$Q_{p/m} = \epsilon n D_{p/m} \mp \sum Q_{losses}, \tag{1}$$

$$D_{p/m} = const, n = variable, \epsilon = 1 \tag{2}$$

where $Q_{p/m}$ is the flow rate thru the pump/motor, $D_{p/m}$ is displacement, *n* is the rotational speed. Leakages in the hydraulic part of the system are described via equation further:

$$Q_{p/m} = \frac{D_{p/m}\Delta p_{AB}}{2\pi\mu} \mp C_S \frac{D_{p/m}\Delta p_{AB}}{2\pi\mu} \mp Q_R, \tag{3}$$

where $C_S$ is a slip coefficient based on laminar flow, $Q_R$ is a constant leakage flow, and $\mu$—fluid dynamic viscosity.

Torque of the pump/motor is a load for EM and it is calculated as

$$T_{p/m} = \frac{D_{p/m}\Delta p_{AB}}{2\pi} \pm C_f \frac{D_{p/m}\Delta p_{AB}}{2\pi} \pm C_V \mu D_{p/m} n \pm T_C, \tag{4}$$

where $C_f$ is a Coulomb friction coefficient, $C_V$ is a viscous friction coefficient, $D_{p/m}$ is pump/motor displacement, and $T_C$ is constant torque loss.

Cylinders model can be described by the following equations:

$$V_A = V_{A,0} + A_A x, \tag{5}$$

$$V_B = V_{B,0} + A_B(x_{max} - x), \tag{6}$$

where $V_{a,o}$ and $V_{a,b}$ are dead volumes consisting of inlet port volume, the volume of the cylinder when the piston is in the end position, and fluid volume in cylinder hoses, $X$ is piston position and $X_{max}$ is cylinder stroke.

Internal leakage is expressed as

$$Q_{leakage} = K_L(p_A - p_B), \tag{7}$$

where $K_L$ is an internal laminar leakage coefficient. Consequently,

$$\frac{\delta p_A}{\delta t} = \frac{K_f}{V_A}(Q_{v.A} - Q_{v.leak} - A_A \dot{x}), \tag{8}$$

$$\frac{\delta p_B}{\delta t} = \frac{K_f}{V_B}(Q_{v.B} - Q_{v.leak} - A_B \dot{x}), \tag{9}$$

where $K_f$ is a bulk modulus of the fluid.

As a result force of cylinder is defined as

$$F_{net} = p_A A_A - p_B A_B - F_{seal} - F_{end} = F_{load} + m\dot{v}, \tag{10}$$

The accuracy of the hydraulic model was increased in comparison with the previous study [1]. This became possible due to the introduction of loss modeling. Moreover, the non-linearized model was addressed and solved this time.

The mathematical model of the PMSM is based on differential equations that describe the torque, current, and voltage behavior of the EM. For more details, refer to the works in [23,24]. As mentioned earlier, the FOC is used as the control system for the PMSM and its structure is presented in Figure 2 below.

The outer loop regulates the speed of the system. The FOC consists of three main control loops with PI controllers. These loops regulate the electromagnetic torque and flux of the EM. The coefficients of all PI controllers are selected or calculated. The outputs of the PI controllers are geometrically transformed from rotating three-phase systems to stationary two-phase systems, which significantly simplifies control. Two vectors from the two-phase system are utilized as inputs of the SVM block. The SVM block is a frequency-based unit, which forms the control signals for the Insulated-Gate Bipolar Transistors (IGBTs) in the FC. A comparison of the PWM-forming strategies was completed in a previous research study [25]. The frequency converter consists of 12 (IGBT) transistors, which are controlled by the SVM.

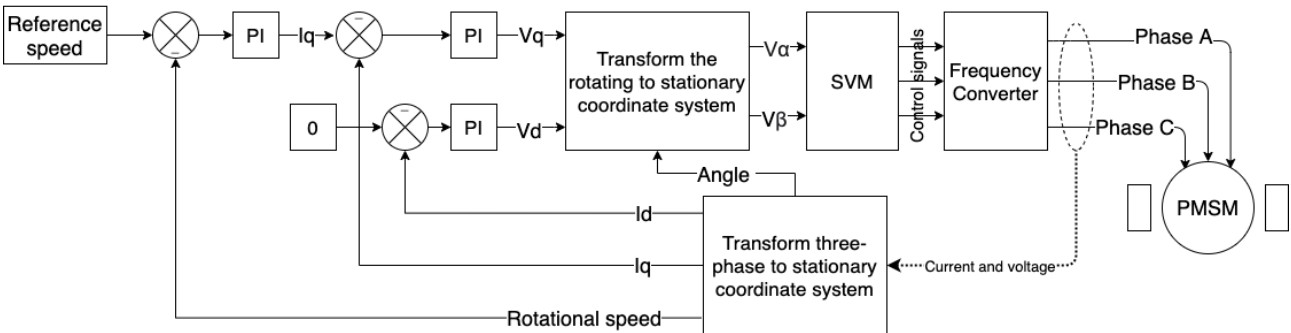

**Figure 2.** Field-oriented control structure.

The whole model of the system is a combination of the hydraulic model and electric drive model. These parts of the system are connected via the torque, rotational speed, and position of the stroke. The validation results of the system are demonstrated in the next section.

## 3. Model Validation

This section describes the validation of the simulated model with the test rig. A photograph of the test rig is shown in Figure 3 below. The structure of the test rig was demonstrated in Figure 1 earlier.

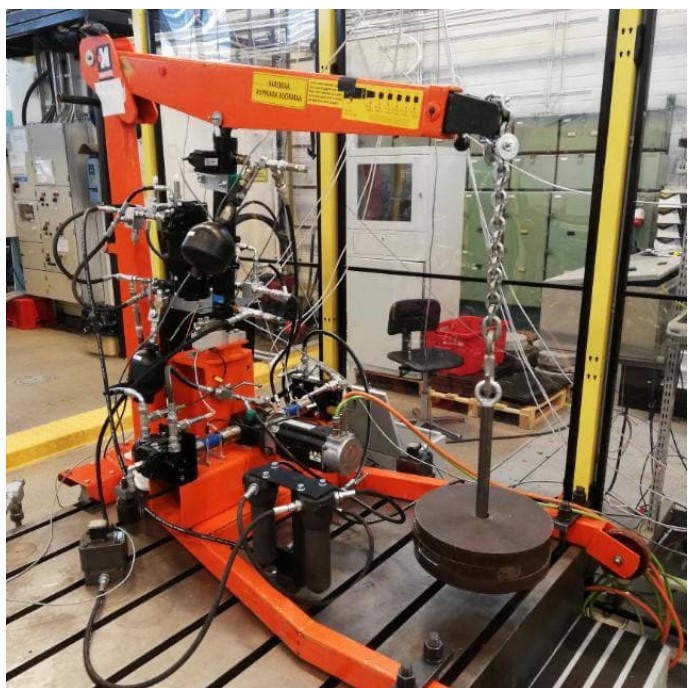

**Figure 3.** Laboratory test rig.

Validation is implemented during the operational cycle of the cylinder (i.e., the extension/retraction sequence). Extension and retraction are implemented by the speed control of the electric drive, which rotates the pump/motor pair. The maximal rotational speed of the electric motor is 500 rpm, and the payload is zero kg. The time to complete one extension/retraction sequence of the electric drive is approximately 21 s. The characteristic of the simulated/experimental rotational speed of the EM is presented in Figure 4.

Both characteristics (simulated and experimental) have a similar behavior during the whole cycle. The difference between the simulated and experimental values is around 10 rpm in both directions of the rotation. However, the simulated characteristic demon-

strated a 1.5 times faster transient process and 2.5 times smaller amplitude of fluctuations in comparison with the test rig.

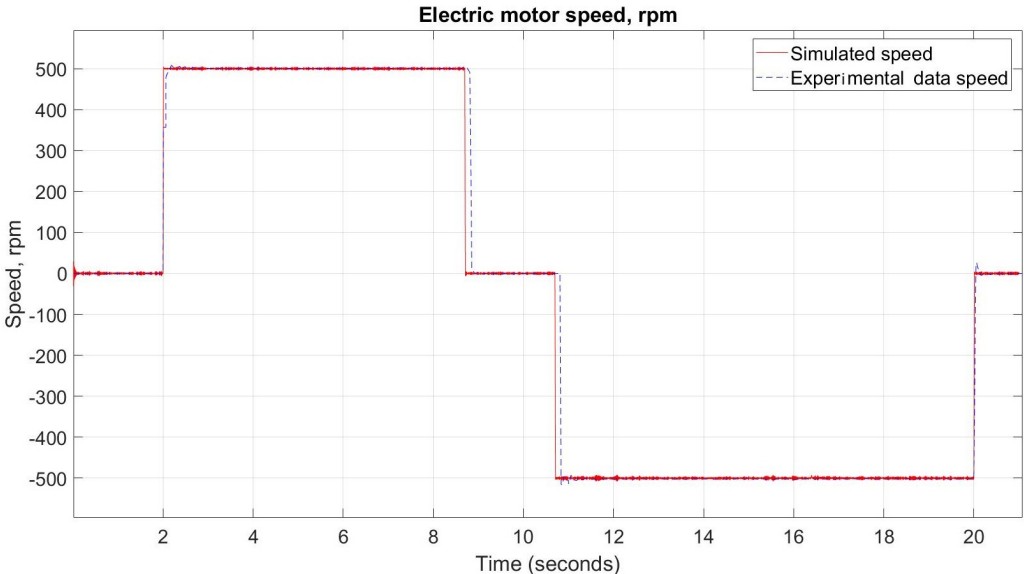

**Figure 4.** EM speed validation.

Changing the characteristic behavior of the flow in the system is the result of the speed regulation of the EM. The simulated and experimental piston side flow of the system is demonstrated in Figure 5. Here, the position of the cylinder is limited by 31 cm. The characteristic of the stroke position is presented in Figure 6 below.

Most of the time, the cycle characteristics are very similar. According to Figure 5, the simulated characteristic has a 3.5% smaller level of fluctuations on the negative flow side and 2.4% on the positive one. Thus, the simulated flow rate graphic reaches the steady-state value faster than an experimental one. Moreover, the experimental flow rate has a reverse flow from 8.9 s to 9.5 s. It happened because the flow rate was not zero after full cylinder extension. As a result, some of the hydraulic fluid gets back before the equilibrium state. Changing the flow in the system allows for the movement of the piston in the cylinder as an actuator.

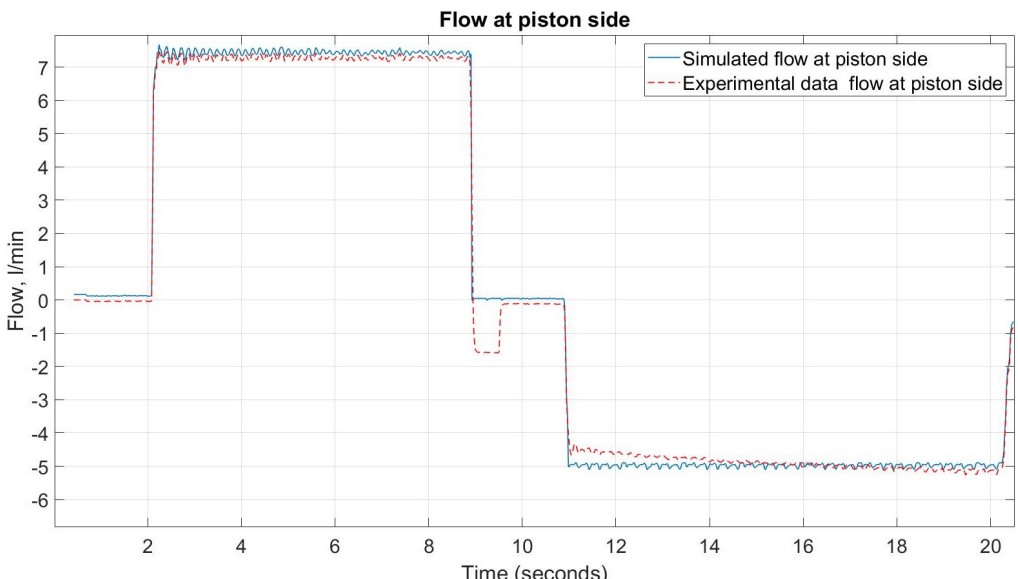

**Figure 5.** Piston side flow validation.

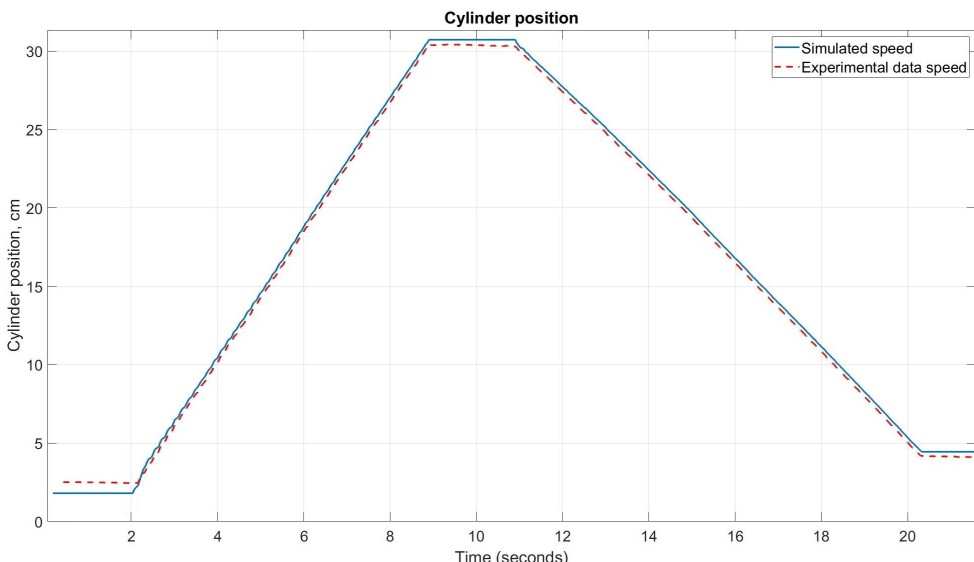

**Figure 6.** Cylinder's position validation.

According to Figure 6, the difference between the simulated and experimental stroke position is around 0.32 cm on average. However, this difference is approximately 0.66 cm at the beginning, because of the initial conditions between the simulation and experimental processes.

Similar characteristics and system behavior were obtained after the simulation of the DDH system with the reference data from the experimental test rig and the usage of the same EM speed control. Simulation errors and differences in values did not exceed 5%. Thus, validation is counted as sufficient, and the model can be utilized instead of a test rig for further research.

## 4. Simulation Study

This section represents the results of the simulation study, which was performed as a part of this research. A step function with no load system response is illustrated in Section 4.1. A PWM analysis under zero payload is presented in Section 4.2. The simulation of the operation of the system under non-zero constant payload is covered in Section 4.3. Information from previous DDH experimental studies [1] was used for fast tuning of the PI controllers of the model.

### 4.1. Stability Analysis

A step-function speed reference signal with an amplitude of 1500 rpm is utilized in the first part of the simulation. The transient process of the simulated system is shown in Figure 7.

According to Figure 7, the DDH system has a fast transient process. This is explained due to the small mass of the motor, a small inertia, and a load torque of less than 7 Nm. Load, inertia, and power do not exceed the rated values (i.e., power equals 2.54 kW). The rise time of speed (i.e., the time required to first reach the reference value) is approximately 0.051 s. The overshoot is equal to 0.53%, which corresponds to the difference between the reference and simulated values, equal to 8 rpm. A characteristic of the transient process is non-periodic with a steady-state value of 0.16% less than the reference one. The parameters of the system transient process are presented in Table 3. Utilization of a higher payload in the process increases the load torque, and as a result, the transient process becomes monotonic. Immediate acceleration of the EM under a high load (similar to the nominal) increases the electromagnetic load of the EM and current. Obviously, the additional payload increases the inertia, which in turn increases overshoot, which leads to a decrease in stability and an overrunning of the system.

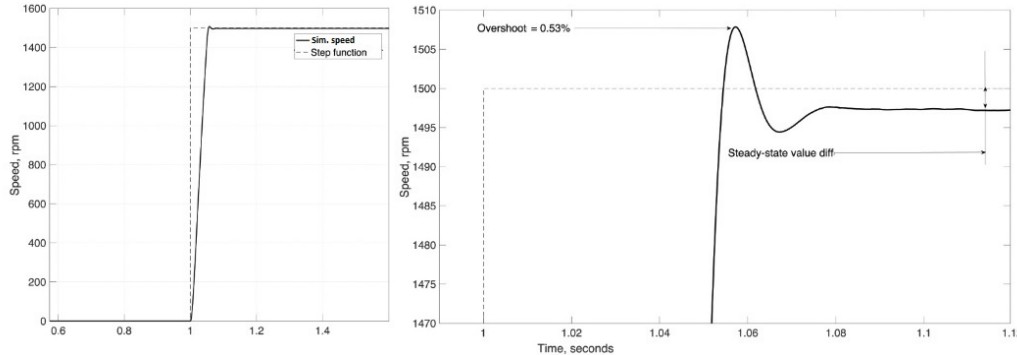

**Figure 7.** Simulated and reference speeds of the system.

**Table 3.** Parameters of step response transient process.

| Rise Time, [s] | Transient Process Time, [s] | Overshoot, [%] | Steady-State Value Difference, [%] |
|:---:|:---:|:---:|:---:|
| 0.051 | 0.13 | 0.53 | 0.16 |

*4.2. PWM Influence with Zero Payload*

This subsection represents the comparison of the system behavior under different PWM frequencies to determine the optimal mode of electric drive operation in terms of the accuracy of speed regulation. Accuracy (e.g., precision) of the speed regulation creates a smaller influence on the hydraulic part behavior. The full scale and zoomed speed of the EM under 5000 Hz are demonstrated in Figures 8 and 9, respectively.

Decreasing the speed fluctuations and overshoot helps to obtain a more stable flow produced by the pumps. As a result, the slight and predictable operation of the actuator should be reached. 10,000 Hz, 7500 Hz, 5000 Hz, 3500 Hz, and 1000 Hz frequencies were utilized for the evaluation. Moreover, the threshold PWM frequency was found for the studied case. This frequency is the minimum frequency that allows the FC to work with minimum overshoot and transient process time values.

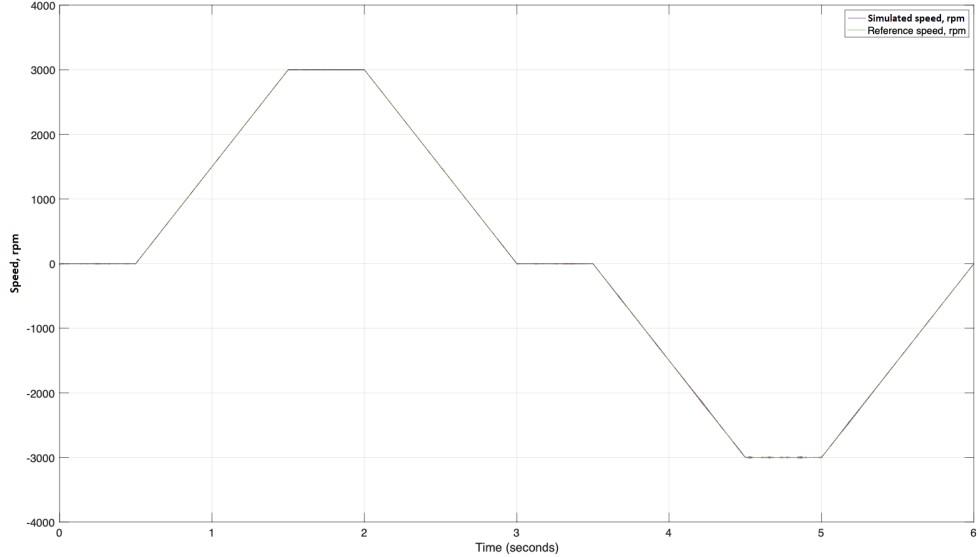

**Figure 8.** Simulated and reference speeds of the system with PWM = 5 kHz.

According to Figure 9, the overshoot is 4.0 rpm (0.133%) at the maximum speed in lifting and 5.5 rpm (0.183%) in the lowering part of the cycle. A comparison of the overshoot values under the different frequencies is shown in Table 4 below.

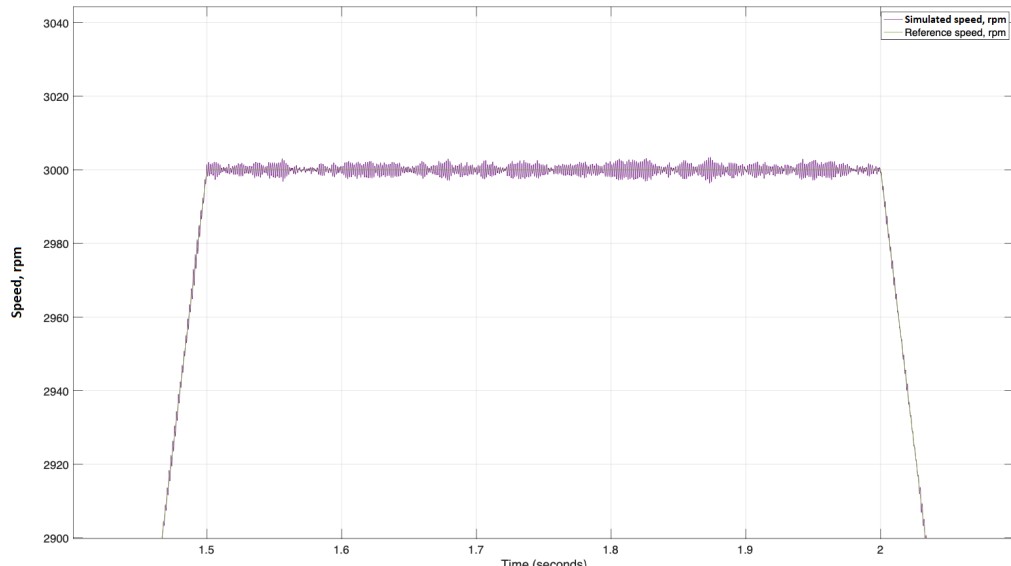

**Figure 9.** Zoomed Simulated and reference speeds of the system (from Figure 8).

**Table 4.** Speed overshoot of EM.

| Frequency, Hz | Lifting | Lowering |
|---|---|---|
| 10,000 | 5.1 rpm/0.17% | 6.4 rpm/0.213% |
| 7500 | 6.3 rpm/0.21% | 7.8 rpm/0.26% |
| 5000 | 4.0 rpm/0.133% | 5.5 rpm/0.183% |

The difference between the lifting and lowering values relies on 5000 Hz as an optimal PWM frequency, which can be explained by the lowest rotational speed overshoot. The EM demonstrates the most accurate speed control (from both sides of rotation, lifting and lowering) in comparison with others. The largest speed error is produced at 7500 Hz, and 10,000 Hz is in the middle. Generally, increasing the PWM frequency improves the behavior of the FC and EM, but the price of the FC grows with an increase in the frequency. However, simplifications in FC control were applied, and the "deadtime" of the IGBTs was not considered. Deadtime is the time required for the reaction of the IGBT. In addition, not all losses were calculated in the FC. All of these factors have an influence on the EM behavior.

The unstable behavior of the system begins from the PWM frequency of 3500 Hz and lower. Operation of the EM under 3500 Hz and 1000 Hz PWM frequencies is illustrated in Figures 10 and 11, respectively.

According to Figure 10, at 3500 Hz, the EM still operates as expected when lifting and is unstable in the lowering part of the cycle. As depicted in Figure 11, at 1000 Hz, the fluctuations of the speed value of the EM exceed 76%, and the operation is unstable in the lowering side as a result of the amplitude fluctuations. Similar characteristics can be expected from the torque (not shown here).

### 4.3. Influence of 20 kg Payload

This subsection describes the behavior of the system in regular conditions. The frequency of the PWM is 5000 Hz. This frequency was chosen because of the smallest value for the error and the low level of fluctuations. Modeling of the hydraulics was improved by the utilization of the nonlinearized differential equations in comparison with the previous studies [1,25]. This modeling utilizes a variable reference speed signal, which was demonstrated in Figure 9 above. A complete working cycle for the model requires 6 s for the lifting and lowering of the cylinder stroke. Acceleration of the EM in this process is

selected in such a way that the currents and electromagnetic torque are in the range of acceptable values (i.e., the overshoot of the transient process must not exceed 20%, and the rise time must be no more than 10% of transient process time). The speed reference for the EM is specially designed for a short movement of stroke without approaching the extreme positions of the cylinder.

The difference between the reference and simulated values of speed is between 2.5 rpm and 4.5 rpm in steady-state mode. This value is better in comparison with the simplified system (i.e., using the linearized hydraulic model) that is utilized in the previous research [1].

Research on the hydraulics operation was performed via several characteristics: stroke position, friction force, and pressure in the chamber of the cylinder. The stroke position and friction force of the cylinder are depicted in Figures 12 and 13, respectively.

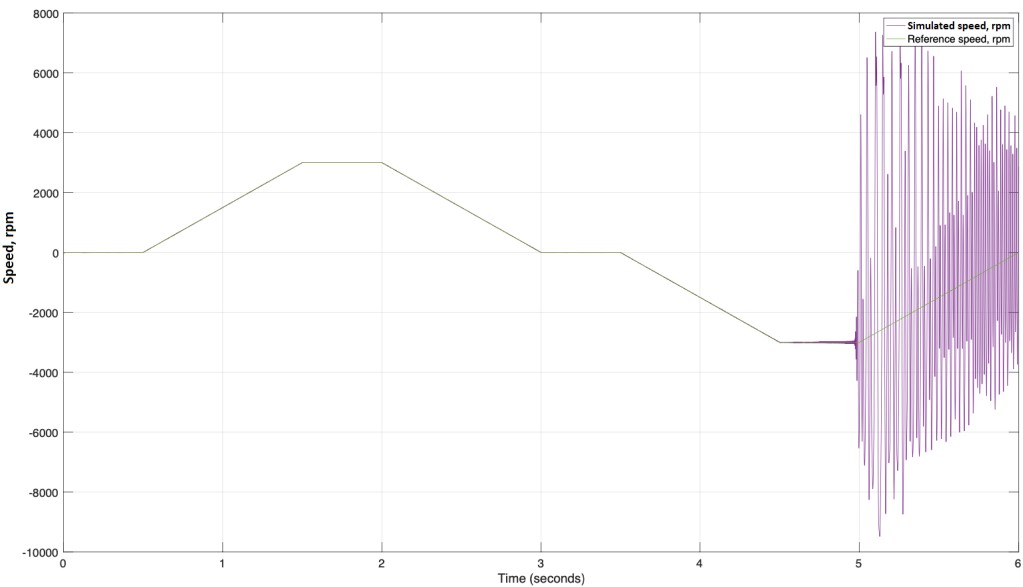

**Figure 10.** Simulated and reference speeds of the system with PWM = 3.5 kHz.

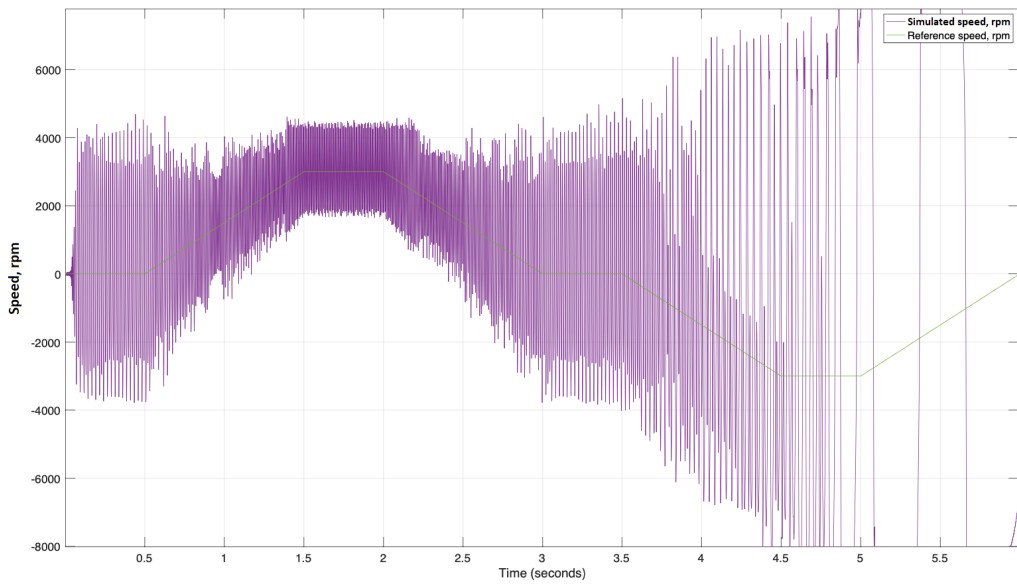

**Figure 11.** Simulated and reference speeds of the system with PWM = 1 kHz.

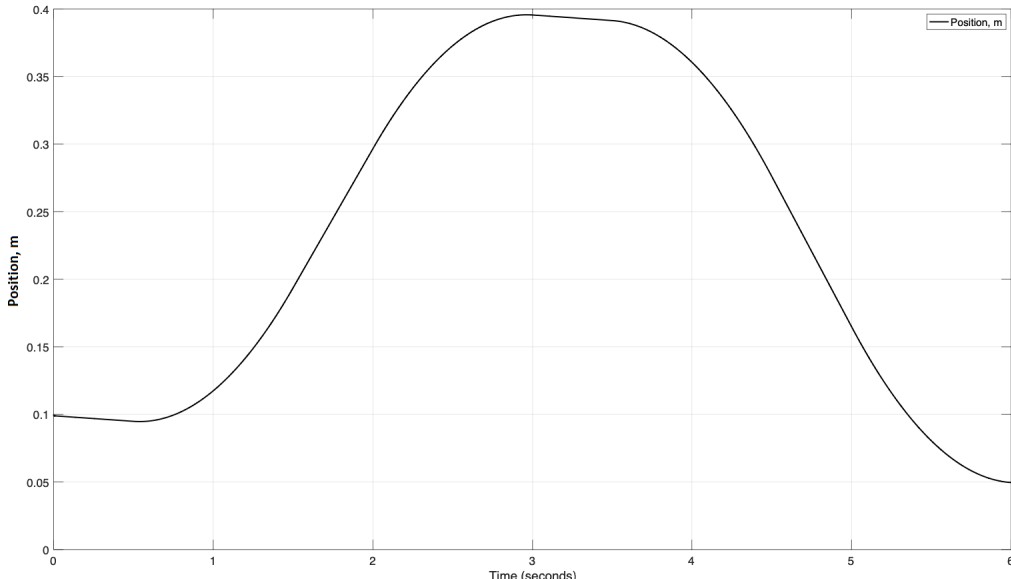

**Figure 12.** Position of the cylinder's stroke.

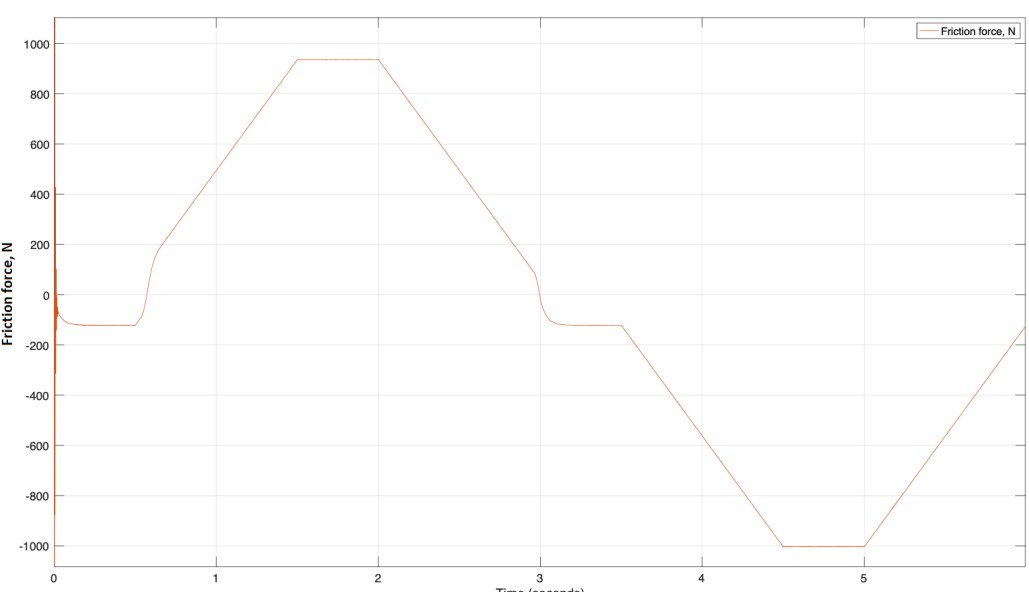

**Figure 13.** Friction force.

The characteristic of the stroke position that is presented in Figure 12 has an asymmetric form. The characteristic takes on the shape of the speed characteristic of the EM with insignificant differences, such as smoother angles and asymmetry. A similar behavior with the rotational speed form is illustrated in the force graph; leakages and other negative forces add nonlinearity to the characteristic.

According to Figure 13, the maximum friction force on the lifting side is 932 N and 1021 N during the lowering. The pressure action of the system is shown in Figure 14; pressure is also quasi-proportional to the rotational speed and has the same friction force shape as the graph.

According to Figure 14, the maximum pressure on the A side (larger) chamber of the cylinder is 7.5 MPa, and the minimum is 6.77 MPa. The friction force and pressure inside the chamber have a leap at the beginning of the simulation. This is explained by the static friction force. To overcome this, the system needs more force and power.

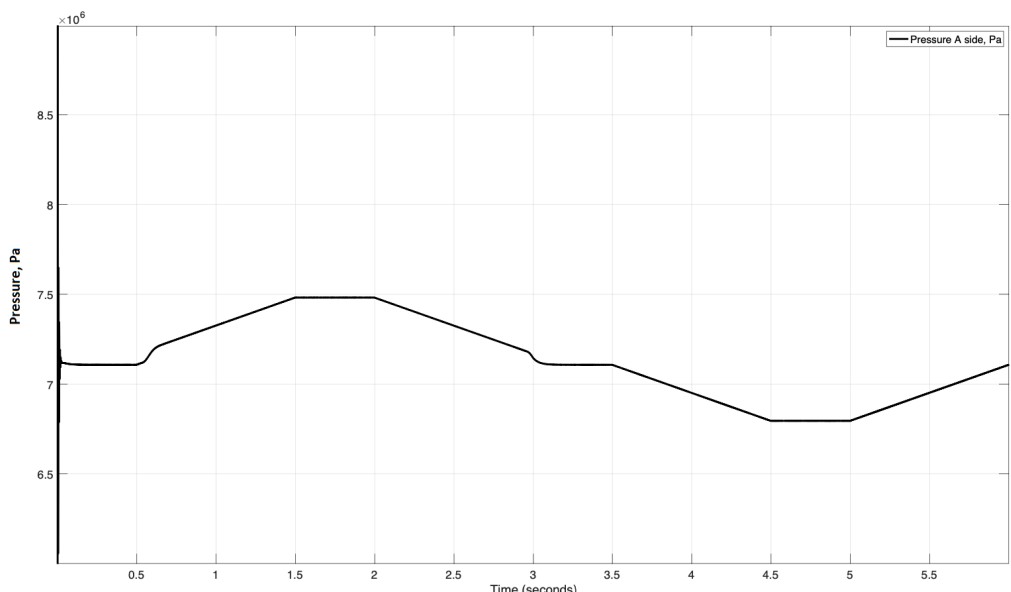

**Figure 14.** Pressure A side.

Apart from the speed evaluation, the torque behavior was also investigated. The action of electromagnetic torque under a 20 kg payload is depicted in Figure 15.

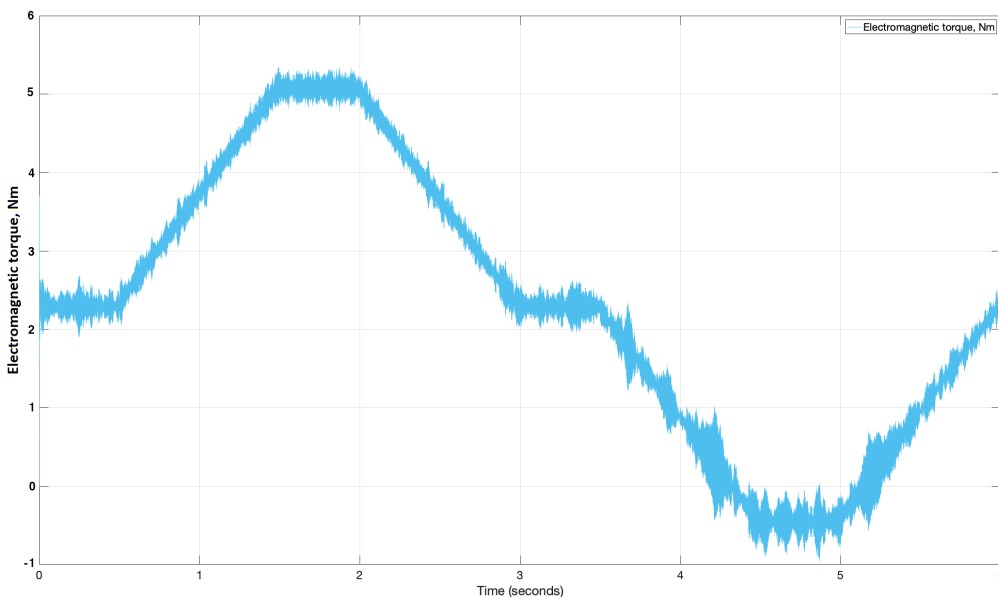

**Figure 15.** Simulated electromagnetic torque of the EM.

Figure 15 shows that the maximum torque amplitude is 5.2 Nm (i.e., less than the rated value of 5.8 Nm). The shape of the graph is close to the reference speed form illustrated in Figure 8; however, the torque characteristic is shifted up. Lowering of the motor requires less electromagnetic torque because of gravity, which helps to retract the cylinder; consequently, the system needs less power and torque. Fluctuations of the torque are 5.77% from the amplitude value in lifting and 11.2% in lowering. The behavior of the consumed current is demonstrated in Figure 16.

The current characteristic can be used to evaluate the amount of consumed energy for the lifting and lowering of the cylinder stroke. A lowering movement from 3.5 s to 6 s consumes less current and power as a result, due to the regenerative abilities of the EM. The average current consumption in the lowering is around 4 times less than in lifting.

In steady-state mode, a Total Harmonic Distortion (THD) level does not exceed 4.57%. The key modeled results for the 20 kg payload case are summarized in Table 5.

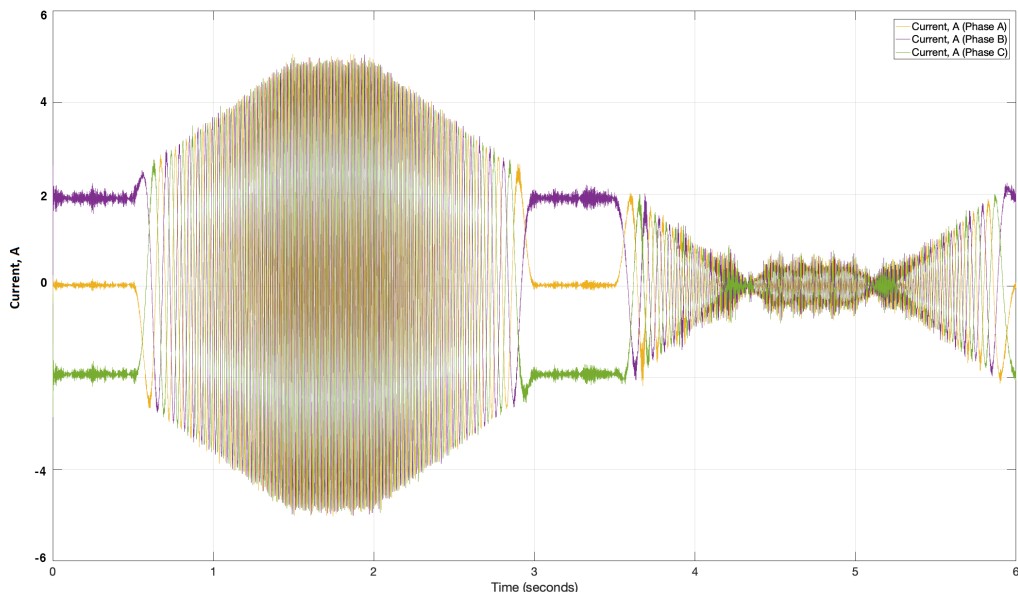

**Figure 16.** Electric motor current.

**Table 5.** Modeling results of operation under 20 kg payload.

| Movement | Sim. Speed Difference, [%] | Torque Fluct., [%] | Friction Force, [N] | Current Ampl., [A] |
|---|---|---|---|---|
| Lifting | 0.05 | 5.77 | 932 | 4.8 |
| **Movement** | **Sim. Speed Difference, [%]** | **Torque Fluct., [%]** | **Friction Force, [N]** | **Current Ampl., [A]** |
| Lowering | 0.05 | 11.2 | 1021 | 2.01 |

Thus, all the simulated values are rather lower than the admissible maximum (in the meaning of control system theory).

## 5. Discussion

During this research, a nonlinearized model of the DDH was built, validated, and simulated (with zero and non-zero) payload. Validation demonstrated that the created model fully corresponds to the laboratory test rig. Accurate positioning within the entire piston stroke demonstrated that the model works as expected. Moreover, the step-function reaction demonstrated that the system is stable. The rise time of the simplified mode is about 0.051 s with an overshoot of 0.53%. The whole transient process time is approximately 0.13 s, and the steady-state value difference is 0.16% in comparison with the reference signal. All these results demonstrate that the system was tuned properly with acceptable behavior in terms of stability.

PWM research demonstrated that frequencies less and equal to 3500 Hz cannot operate in a stable manner. At these frequencies, the stable behavior of the system was observed only during the extension of the cylinder. In particular, the 1000 Hz frequency demonstrated huge fluctuations (76%) and unstable behavior. Frequencies above 5000 Hz work properly; however, the 5000 Hz mode demonstrated the best accuracy and the lowest level of fluctuations (4.0 rpm/0.133% in lifting and 5.5 rpm/0.183% in lowering). As a result, the electric drive is optimized for accurate speed regulation (i.e., minimum overshoot and fluctuations) and further used in the electro-hydraulic system.

Operation under a 20 kg payload demonstrated that the simulated speed loss is 0.05% in comparison with the referred speed under the same payload. During simulation, fluctuations in the electromagnetic torque behavior were 5.77% in the lifting part of the cycle and 11.2% in the lowering part of the cycle. This can be explained by the vertical placement of the cylinder. The reverse movement of the actuator allows it to work in regenerative mode, but it was not implemented in this research. Changing the direction of the movement has an influence on the control system (i.e., increases the computational load) and creates additional disturbances. Moreover, it increases the time of the transient process, overshoot, and additional fluctuations of speed and torque. Force, which is applied to the stroke of the cylinder, has the same direction as the stroke movement. The friction force had a substantial rise at the beginning of the motion, which is explained by the static friction. The maximum friction force is 1021 N and 932 N in lowering and lifting directions, respectively. The friction force behavior can be evaluated as an expected one. As well, the consumed current did not exceed 4.8 A, which fits within the nominal functioning of the motor.

Nevertheless, further work is required. The PWM investigation does not consider the "deadtime" of the IGBTs. The torque fluctuations can be decreased by additional signal filtering. More experimental tests with higher payload variety are needed. Moreover, the harvesting energy mode needs to be implemented to increase usability.

### 6. Conclusions

In this research, the DDH model was built, validated with the experimental test rig, and investigated with a PMSM model controlled by a FOC in Matlab/Simulink. Validation of the system with zero payload and an EM speed of 500 rpm demonstrated a similar behavior with the experimental test rig. All the parameters of the simulated model did not exceed experimental data by more than 5%. A step-function response with various payload simulations was used for the evaluation of EM behavior and hydraulic systems. An acceptable level of stability and accuracy was achieved. The step-function response demonstrated a rise time smaller than required. The overshoot of the system did not exceed 0.53%, and the difference between the reference and simulated speeds in steady-state mode is 0.16%. Furthermore, a PWM analysis demonstrated that the 5000 Hz frequency is optimal, which yields the highest degree of stability, the smallest difference between reference and operated speed, the smallest level of torque fluctuations, and stable and accurate flow control through the pump/motors. Smaller frequencies have worse stability, and higher frequencies have less accuracy. The 10 kHz frequency of the PWM has worse accuracy than 5 kHz but better than 7.5 kHz. This behavior is detected in either the lifting or lowering operations. Normally, there is a trend of improving the behavior of EM by increasing the PWM frequency. However, nonlinear dependence is observed under the hydraulic load. Moreover, there is a threshold value of 3500 Hz. At this point, the observed phenomena cannot be explained, and further investigation is needed. Simulation of the operation of the system under a 20 kg payload demonstrated a speed difference between the reference and simulated speed in steady-state mode of 0.05% and torque fluctuations of less than 12%. Fluctuations have a decaying character. To sum up, this research demonstrated the key features of the behavior of an electric drive within DDH and pointed out the next development steps. The obtained results are satisfactory and are expected to be improved upon in further research.

**Author Contributions:** Conceptualization, V.Z. and T.M.; methodology, V.Z. and T.M.; software, V.Z.; writing—original draft preparation, V.Z.; writing—review and editing, T.M.; supervision, T.M., project administration, T.M.; funding acquisition, T.M. All authors have read and agreed to the published version of the manuscript.

**Funding:** This work was enabled by the financial support of Academy of Finland (project ESTV) and internal funding from the Department of Automation Technology and Mechanical Engineering, IHA group at Tampere University, Finland.

**Conflicts of Interest:** The authors declare no conflict of interest.

**Abbreviations**

The following abbreviations are used in this manuscript:

AC        Alternative Current
DC        Direct Current
DDH       Direct-Driven Hydraulics
EM        Electric Motor
FC        Frequency Converter
FOC       Field-Oriented Control
ICE       Internal Combustion Engine
IGBT      Insulated-Gate Bipolar Transistor
NRMM      Non-road mobile machinery
PMSM      Permanent Magnet Synchronous Machine
PWM       Pulse Width Modulation
THD       Total Harmonic Distortion

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
