# Peer review of "Influence of Hydraulics on Electric Drive Operational Characteristics in Pump-Controlled Actuators"

_actuators, doi:10.3390/act10120321_

Round 1

Reviewer 1 Report

Dear authors, my opinion is that the work is interesting but the discussion of results has to be improved.
While the context of the work is clear, can you add a deeper discussion on the results, a part from the numbers obtained, going more inside "the why and what" if and where is possible?
Right now the work looks more as a technical report rather than a paper. 
Moreover, I've attached the pdf file where I've hightlighted the points not clear to me and some minor errors to be adjusted.

Author Response

Dear reviewer, all the responses to comments are attached in PDF file. PDF also consists of changes in the paper. Please see the attachment.

Best regards, Viacheslav Zakharov.

Reviewer 2 Report

In this article, a simulation model of direct-driven hydraulic system is presented. The direct-driven hydraulic system has many benefits in compare with traditional hydraulic system and present very interesting area for scientific research. I encourage the authors to continue their scientific research of direct-driven hydraulic system, but I reject this article because the research not conducted correctly.

In this article, the authors compare their simulation results with previous research results and they concluded that the new results are better. But that is not appropriate scientific method for the results validation. The simulation results must be compared with the results obtained by experimental measurement on a real or laboratory hydraulic system. The developed numerical model in MATLAB/Simulink and presented simulation results don’t represent a scientific contribution without comparison with real direct-driven hydraulic system. Also, in some part of article the authors used identical text as in previously published papers.

Further, at figure 1. the symbol for the hydraulic pump/motor component is not correct, it is symbol for the hydraulic motor. I advise the authors to have a care about basic knowledge of hydraulic when they publishing scientific article.

Author Response

(The authors gave the same response as above.)

Round 2

Reviewer 1 Report

Only one of the authors suggested has been chosen and introduced in the literature review, but withouth an explanation on the rebuttal letter. (I mean: why only that one?at least explain me that) Anyway, after suggesting to expand the literature one time, I don't feel I can force the authors to do so. I'm a little bit sorry about it, I suggest in the future the authors should care more and answer the concerns of reviewers with more expanded explanations. That's why I've selected "can be improved" for the introduction section.

Reviewer 2 Report

The authors have made a major revision of the paper, so I agree to the publication of this paper.